# Green Synthesis of Fe_x_O_y_ Nanoparticles with Potential Antioxidant Properties

**DOI:** 10.3390/nano12142449

**Published:** 2022-07-17

**Authors:** Johar Amin Ahmed Abdullah, Mercedes Jiménez-Rosado, Víctor Perez-Puyana, Antonio Guerrero, Alberto Romero

**Affiliations:** 1Departamento de Ingeniería Química, Escuela Politécnica Superior, Universidad de Sevilla, 41011 Sevilla, Spain; mjimenez42@us.es (M.J.-R.); aguerrero@us.es (A.G.); 2Departamento de Ingeniería Química, Facultad de Química, Universidad de Sevilla, 41012 Sevilla, Spain; vperez11@us.es

**Keywords:** iron oxide nanoparticles, green synthesis, *Phoenix dactylifera* L., polyphenols, antioxidant activity

## Abstract

Iron oxide nanoparticles (Fe_x_O_y_-NPs) are currently being applied in numerous high-tech sectors, such as in chemical sectors for catalysis and in the medical sector for drug delivery systems and antimicrobial purposes, due to their specific, unique and magnetic properties. Nevertheless, their synthesis is under continuous investigation, as physicochemical methods are considered expensive and require toxic solvents. Thus, green nanotechnology has shown considerable promise in the eco-biogenesis of nanoparticles. In the current study, Fe_x_O_y_-NPs were synthesized by two different methods: via green synthesis through the use of polyphenols, which were extracted from *Phoenix dactylifera* L.; and via chemical synthesis, in which the reducing agent was a chemical (NaOH), and iron chloride was used as a precursor. Thus, polyphenol extraction and its ability to produce nanoparticles were evaluated based on the drying temperature used during the *Phoenix dactylifera* L. recollection, as well as the extraction solvent used. The results highlight the potential of polyphenols present in *Phoenix dactylifera* L. for the sustainable manufacture of Fe_x_O_y_-NPs. Finally, green and chemical syntheses were compared on the basis of physicochemical characteristics and functional properties.

## 1. Introduction

Nanomaterials (1–100 nm in size) are gaining considerable interest for their unique properties. Thus, they can be used in several sectors and applications, with continuous growth [1,2,3]. These nanomaterials can be obtained by various methods, which are classified into two types: natural nanomaterials, such as biogenic magnetite; and anthropogenic nanomaterials, which include all manufactured nanomaterials. These nanomaterials are generally labelled based on their shapes and structures (e.g., nanoparticles, nanotubes, nanofibers. etc.) [4,5]. Nevertheless, most of the physicochemical methods used to develop nanomaterials require the use of expensive and toxic solvents [6,7,8]. The use of environmentally friendly methods (green methods) to synthesize nanoparticles from noble metals has recently been investigated, with the aim of providing high-purity NPs through simple, economic and eco-friendly techniques [9,10,11].

For this purpose, plant and fruit extracts, as well as bio-organisms, have been used to achieve environmentally safe methods [12,13,14,15,16]. The importance of these green methods is not only based on environmental factors but also on other interesting features that they may provide. Thus, it has been shown that green methods are typically inexpensive, effective, rapid and non-toxic, generally producing nanoparticles with high crystallinity and various shapes and sizes [17,18,19]. These characteristics were reported to depend on two groups of parameters, including the composition of raw materials and additives used, as well as the processing conditions applied during the synthesis of NPs. Among compositional parameters, the most important are the nature and relative concentration of the plant extract (i.e., natural polyphenol content), the metal salts used as precursors, the extract/metallic salt ratio, the pH, and the addition of a reducing agent. The most important processing conditions are the drying temperature, grinding particle size and the extraction method employed, in addition to reaction temperature and time [20,21]. It has also been reported that the functionality of these extracts comes from their high polyphenol ratios, which reduce metal salts into high-purity nanoparticles due to their unique properties (reducing character, ability to form hydrogen bonds, nucleophilic activity, polarizability, acidity and chelation ability) [22,23].

As a suitable candidate for this purpose, *Phoenix dactylifera* L. is a rich source of polyphenol content. Its use was reported as an antioxidant additive in antimicrobial, antilipemic and antidiabetic activities [24]. Thus, its use in the synthesis of nanoparticles, particularly Fe_x_O_y_-NPs, could confer magnetic properties, high electrical conductivity, high-temperature properties and suitable biocompatibility, enabling them to be used in biomedical applications, such as drug delivery, antimicrobial treatments and diagnosis [25,26,27,28,29,30]; in chemical applications, such as energy conversion, [31] catalysis [32] and environmental protection [33,34]; in electronic and optoelectronic applications [35]; and in agriculture and biotechnology applications in animals [36,37,38,39,40].

In order to maximize the polyphenol content in extracts, it is vital to understand the factors that determine polyphenol content in each processing stage, as well as the effect of phenolic content on the physicochemical characteristics of Fe_x_O_y_-NPs. It is known that the development of the phenolic content is modulated by pre-harvesting factors. These include intrinsic factors of the plant (e.g., *Phoenix dactylifera* L.), such as genetics and age, as well as environmental factors, such as exposure to solar radiation, rainfall and soil type [7]. In addition, the assessment of post-harvest parameters related to the processing and storage of the plant is also considered important (e.g., drying method, grinding, extraction method, solvent effect, extraction time and extraction temperature) [7,8,41,42]. According to some studies, drying temperature, grinding time and solvent type have a direct correlation with the phenolic compound extracted [8,43]. Various solvent systems were used to extract phenolic compounds from plants [44]. However, it was found that the type of extraction is the most important factor influencing the yield and antioxidant properties of polyphenols. This was associated with the abundance of antioxidant compounds with diverse polarity and different chemical properties, which make them solvent-soluble or not [45]. Phenolic content could have an important influence on some of the physical characteristics of nanoparticles, although information related to this influence is very limited [46]. A few authors have reported the possibility of establishing a direct relationship between phenolic compound concentration and Fe_x_O_y_-NPs size and type using plant extracts [46,47]. Specifically, the novelty of the present work lies in evaluating the influence of the polyphenol properties of *Phoenix dactylifera* L. on the synthesis of Fe_x_O_y_-NPs and their characteristics, such as their size, crystallinity, type and morphology. The previously mentioned studies did not provide information regarding the influence of phenolic compounds on the type and portions of Fe_x_O_y_-NPs, crystallinity degree or possible synthesis mechanisms. For example, in a 2019 study, Salgado et al. [46] reported a mixture phase of iron oxide nanoparticles (Fe_x_O_y_-NPs) from various extracts, modifying the reaction pH to 6 by adding 1 M of NaOH. There might be a further impact of the pH of the reaction on the phenolic compounds in the extracts, thereby affecting the Fe_x_O_y_-NPs. It should be noted that there was no modification to pH in this experiment, which allows for attribution of the actual impact of phenolic compounds present in the extract to those compounds.

The main objective of this study was to synthesize Fe_x_O_y_-NPs through green synthesis using *Phoenix dactylifera* L. extracts. To this end, extracts were first collected using different drying temperatures (25, 50, 100 and 150 °C) and extract solvents (water and ethanol). Nanoparticles were also synthesized by a chemical method to compare them with green NPs. The comparison took into account the physicochemical characteristics of the nanoparticles and their antioxidant activity (TAC and DPPH).

## 2. Materials and Methods

### 2.1. Materials

Folin–Ciocalteu reagent, sodium carbonate (Na_2_CO_3_), methanol (CH_3_OH), ethanol 99.9%, iron (III) chloride hexahydrate 98% (FeCl_3_·6H_2_O), sodium hydroxide (NaOH), sulfuric acid (H_2_SO_4_), sodium phosphate (Na_2_HPO_4_), ammonium molybdate ((NH_4_)_3_PMo_12_O_40_), gallic acid (C_7_H_6_O_5_), hydrochloric acid (HCl), 2,2-diphenyl-1-picrylhydrazyl (DPPH) and dimethyl sulfoxide anhydrous (DMSO) 99.9% were provided by Sigma Aldrich (Darmstadt, Germany). The rest of the reagents employed in this work were of analytical grade.

*Phoenix dactylifera* L. leaves were obtained directly from trees from Seville (Seville, Spain), where the mean temperature is 19 ± 4 °C (with minimum and maximum temperatures of 6 °C and 36 °C, respectively) and the average relative humidity is 53%, ranging between 33 and 73% (data from the National Oceanic and Atmospheric Administration, Washington, DC, USA).

### 2.2. Methods

Figure 1 shows a diagram of the process carried out to obtain NPs from plant sources. The polyphenols were first extracted from *Phoenix dactylifera* L. leaves, and later, nanoparticles were produced by green (GS) or chemical (CS) synthesis through colloidal precipitation.

#### 2.2.1. Polyphenol Extraction from *Phoenix dactylifera* L. Leaves

First, *Phoenix dactylifera* L. leaves were thoroughly washed three times with distilled water and dried at different temperatures (25, 50, 100 and 150 °C) until the water loss percentage was constant. The drying kinetic parameters were determined by the transformation of moisture content into water loss percentage and applying a nonlinear curve (Gompertz fitting function), according to Equation (1):(1)$$WL\left(t\right)=WL\left(\infty \right)\cdot e^{-e^{\left(1-k\cdot t\right)}}$$
where WL(t) is the percentage of water loss over time (t)*,*
WL(∞) is the maximum percentage of water loss when t → ∞ and k is the drying rate constant (i.e., the rate constant of water loss percentage per time unit) [48].

Then, the samples were pulverized with an electric grinder (Cuisinart Grind Central DCG-12BC, East Windsor, NJ 08520 Printed in China) and sieved to obtain a fine powder. Subsequently, the powders were characterized using a Mastersizer 2000 (Malvern Instruments Ltd., Malvern, UK) in order to determine the grain size distributions (µm) in the dried samples. 

Next, 30 g of each powder was extracted using a Soxhlet extractor employing 300 mL of distilled water or ethanol 99.9% as solvents at 98 or 68 °C, respectively, for 8 h. The solvent was recovered, and the yield of extraction was calculated using Equation (2) [49]: (2)$$y=\left(\frac{Mass\;of\;\;crude\;extracted\;}{Inicial\;mass\;of\;phoenix\;powder}\right)\times 100$$

The crude extracts were dissolved again in water or ethanol and centrifuged at 10,000 rpm for 15 min. Finally, the supernatant (extract) was collected and stoked at 4 °C for further use.

#### 2.2.2. Total Polyphenolic Content (TPC)

To determine the total polyphenol content (TPC) in *Phoenix dactylifera* L. extract, we used the method described by Ayala-Zavala et al. [50] and Zapata et al. [51]. According to this method, 50 µL of the extract was mixed with 3 mL of distilled water and 250 µL of the Folin-Ciocalteu 1N reagent until equilibrium was reached (8 min). After mixing, 750 µL of 20%, Na_2_CO_3_ and 950 µL of distilled water were added. Finally, the absorbance was measured in a UV/VIS spectrophotometer (PG Instruments Ltd., model T70 + UV/VIS Spectrometer) at 765 nm after 30 min of incubation at room temperature. A calibration curve was also prepared for gallic acid. The results were expressed in mg gallic acid equivalents per gram of *Phoenix dactylifera* L. extract (mg GAE/g of extract).

#### 2.2.3. Nanoparticle Preparation

##### Green Approach

In this approach, the GS FexOy-NPs were synthesized by mixing 20 mL of each extract (0.27 g/mL) with 20 mL of FeCl_3_·6H_2_O 1 M solution for 2 h at 50 °C. Then, the mixture was filtered using filter paper (Whatman n°1) and washed at least three times with distilled water to ensure that any impurities and foreign particles suspended in the mixture were removed. The filtered solid was then kept at 100 °C for 8 h and, subsequently, hot-calcined at 500 °C for 5 h to obtain the purest nanoparticles possible.

##### Chemical Approach

Nanoparticles were also synthesized chemically by mixing 20 mL of NaOH 1 M with 20 mL of FeCl_3_-6H_2_O 1M. The remaining steps were the same as those described for the green approach.

### 2.3. Characterization Techniques

#### 2.3.1. X-ray Diffraction (XRD)

Both green and chemically synthesized nanoparticles (GS Fe_x_O_y_-NPs and CS Fe_x_O_y_-NPs, respectively) were subjected to XRD characterization (Brand: Bruker Model D8 advance A25 with Cu anode) to determine the crystal phase composition of the samples. These tests allowed for calculation of the crystallinity index and the crystal size.

The most intense peaks of each XRD pattern were used to determine the crystallite size, applying the formula of Debye-Scherrer [52] (Equation (3)):(3)$$D=\frac{k\lambda }{\beta \cdot cos\theta }$$
where D is the diffracting domain size, k is a correction factor (0.94), *λ* is the used wavelength (0.154178 nm), *β* = FWHM (full width at half maximum) and θ is the position of the main peak.

The crystallinity degree (%) of the GS and CS Fe_x_O_y_-NPs was calculated using Equation (4): [53]
(4)$$\mathrm{Cristanillity}\;\left(\%\right)=\frac{\mathrm{Area}\;\mathrm{of}\;\mathrm{cristalline}\;\mathrm{peaks}}{\mathrm{Cristalline}\;\mathrm{peaks}\;\mathrm{area}\;+\;\mathrm{amorphous}\;\mathrm{area}\;}\times 100$$

#### 2.3.2. Fourier Transform Infrared Spectroscopy (FTIR)

FTIR was carried out to obtain information about the vibration modes of the bonds present in Fe_x_O_y_-NPs and their structure.

The GS and CS Fe_x_O_y_-NPs and Phoenix dactylifera L. raw powder were subjected to FTIR spectroscopy (Nicolet iS50 FITR Spectrometer, ThermoFisher Scientific, Madison, WI, USA) from 4000 to 400 cm^−1^ with a resolution of 0.482 cm^−1^ to identify the bonds present in the nanoparticle structure.

#### 2.3.3. Scanning Electron Microscopy (SEM)

The morphology of both GS and CS Fe_x_O_y_-NPs, as well as their diameter distribution average, were measured by SEM with a Zeiss EVO SEM (Pleasanton, CA, USA) at 10 kV, recording images at different magnifications. The images were analyzed with ImageJ software (1.53q; National Institutes of Health, Bethesda, Maryland, USA) (free software) [54].

#### 2.3.4. Determination of Antioxidant Activity

Normally, multiple methods are used to evaluate antioxidant activity due to its complexity and the diverse groups that can present antioxidant activity [55]. Therefore, two distinct methods were used for the extracts and GS and CS Fe_x_O_y_-NPs: total antioxidant activity phosphomolybdate test (TAC) and DPPH antioxidant activity.

Total Antioxidant Activity (TAC) Determination.

TAC analysis was performed following a previously described protocol [56] with a slight modification: 2 mL of sample was mixed with 2 mL of reagent (4 mM (NH_4_)_3_PMo_12_O_40_, 0.6 M H_2_SO_4_ and 28 mM Na_2_HPO_4_). These mixtures were incubated for 90 min at 95 °C in a water bath. Once tempered, the absorbance was determined by spectrophotometry at 695 nm, using gallic acid as a reference. Before the test, the NPs were dispersed in HCl.

DPPH Antioxidant Activity Test

This evaluation was performed using the protocol described in a previous study with a slight modification: a series of nanoparticle dispersion was prepared (4, 2, 1, 0.5, 0.25 and 0.125 mg/mL in DMSO), and subsequently, volumes of 2 mL of these dispersions were mixed with 2 mL DPPH. These mixtures were incubated for 30 min, and later, their absorbance was tested at 517 nm. The control solution was made up of 2 mL DPPH solution and 2 mL of DMSO. The DPPH activity of the extracts was evaluated out using the same protocol [56].

Finally, the inhibition percentage *IC (%)* of each concentration was established (Equation (5)) through the relationship between the absorbance values of the oxidized solutions in the absence of any antioxidant agent and those in the presence of antioxidant agents (*A_D_* and *A_Da_*, respectively).
(5)$$IC\left(\%\right)=\left(\left(A_{D}-A_{Da}\right)/A_{D}\right)\cdot \;100$$

In addition, the necessary antiradical to cause a 50% inhibition (*IC_50_*) of each NP was determined using GraphPad Prism 9 software (GraphPad Prism version 9.0.0 for windows, San Diego, CA, USA, www.graphpad.com) [56].

### 2.4. Statistical Analysis

Statistical analyses were performed using IBM SPSS statistics 26 software and GraphPad Prism9 software. A one-way ANOVA was carried out to estimate the significant differences between observations. In at least three replicates, the data were expressed as mean ± SD. In addition, HSD Tukey tests were conducted to determine the significance level (*p* < 0.05).

## 3. Results and Discussion

### 3.1. Characterization of Raw Material and Extracts

#### 3.1.1. Effect of Drying Temperature on Moisture Content in *Phoenix dactylifera* L.

Figure 2 shows the kinetics of moisture loss in *Phoenix dactylifera* L. The moisture content was converted to water loss percentage from the samples and fitted to a non-linear curve (Gompertz function) (R^2^ ≥ 0.99). The maximum moisture loss percentage obtained was in the following order with significant differences (*p* < 0.05): T_150 °C_ > T_100 °C_
≈ T_50 °C_ > T_25 °C_. The quickest sample was T_150°C_, reaching WL(∞) = 55.7 ± 1.7% (maximum moisture loss) in 8 h, whereas the slowest sample was T_25 °C_, which required more time (1056 h) to reach equilibrium in WL(∞) = 51.1 ± 1.2%. Samples T_50 °C_ and T_100 °C_ did not show any significant difference with respect to WL(∞), with values of 53.8 ± 2.6 and 54.2 ± 1.6% reached in 72 and 48 h, respectively, and drying rate values of *k* = 0.10 ± 0.02 and 0.22 ± 0.03 (H_2_O h^−1^). This indicates that accelerated dehydration occurred in the samples when the drying temperature was increased. In this way, T_150 °C_ reduced the time by 132, 9 and 6 times faster than T_25 °C,_ T_50 °C_ and T_100 °C_, respectively, reaching a higher moisture loss. Similar results were reported by Larrauri et al. [57]. This disposal could be elucidated by the increased drying rate produced by higher temperatures. Thus, it was observed that when the drying temperature was increased, the drying rates (kinetic parameter *k*) increased significantly, as shown in Figure 2B. Similar results were reported by Patrón-Vázquez et al. [48] and Djaeni et al. [58].

#### 3.1.2. Drying Temperature, Grinding and Total Polyphenol Content Effect

Table 1 shows the size average of powdered particles, extraction yield and total polyphenol content (mgGAE/g extract) as a function of drying temperatures (25, 50, 100 and 150 °C) and solvents (H_2_O or ethanol). The particle size was the smallest (36.98 ± 1.07 µm) at T_150 °C_, increasing at lower drying temperatures. This effect may be a consequence of the fact that high temperatures favor mobility, facilitating the extraction of substances that may be present inside the leaf, giving rise to smaller particles after a grinding stage. The extraction yield increased with increased drying temperature. This increased extraction may also be attributed to alterations of internal components of cells induced at higher temperatures, which may also cause these cells to lose a considerable amount of their active substance, regardless of the high yield achieved [8,59,60]. These effects are in line with the results of previous studies [60], alongside the effect of the solvents. Among the tested solvents, ethanol achieved a higher extraction yield than water at all the studied temperatures, except at the highest temperatures, at which no significant differences were found. Nevertheless, all the samples achieved an extract yield of more than 50%, in agreement with the results of previous studies. These results were compared with those of other Soxhlet extracts [49,60].

The total polyphenolic compounds (TPC) and antioxidant activities were also evaluated (Table 1). The results indicate that drying temperature also affects the TPC. In this sense, the lower the drying temperature, the higher the TPC. Thus, the highest TPC was recorded at 25 °C (42.28 ± 0.79 mg GAE/g extract and 38.64 ± 1.72 mg GAE/g extract for water and ethanol, respectively). However, this effect may be considered only a moderate trend up to a drying temperature of 100 °C. At the highest drying temperature, at which the aforementioned loss of active substances from the cells should be relevant, the reduction in TPC was clear for the two solvents used. In addition, a higher TPC was observed in hot water as compared with the organic solvent, similar to the effect on TPC reported in other studies on plant extracts [41,59,61,62]. However, natural phenols have been reported to show a solubility preference for solvents with intermediate polarities, such as acetones and alcohols, rather than more or less polar solvents (e.g., water and ethyl acetate, respectively) [41]. According to Bhebhe et al. [41], an elevated water temperature may also have contributed to a higher TPC, applying the empirical rule previously mentioned, i.e., “like dissolves like”. This could indicate that *Phoenix dactylifera* L. contains more water-soluble polyphenolic compounds and phenolic acids, which have a solubility preference for water, unlike other solvents, due to their lower activity coefficient in water [63].

### 3.2. Characterization of Nanoparticles

#### 3.2.1. X-ray Diffraction (XRD)

Figure 3 shows the XRD diffractograms of GS-Fe_x_O_y_-NPs synthesized from the ethanolic and aqueous Phoenix extracts prepared from powders dried at different temperatures. The XRD diffractogram of CS-Fe_x_O_y_-NPs was also used for the sake of comparison. All the peaks observed in the GS Fe_x_O_y_-NPs diffractograms are indicative of a spinel structure. Thus, the peaks observed at 2θ (°) = 24.15, 33.15, 35.63, 40.86, 49.49, 54.07, 56.16, 57.45, 62.44 and 64.00 can be assigned to the crystallographic reflection planes (012), (104), (110), (113), (024), (116), (211), (122), (214) and (300), respectively, which correspond to the crystalline rhombohedral structure of hematite (Fe_2_O_3_); space group: R-3c N°: 167 with standard crystallographic parameter a = b = 5.035 Å, c =13.751 Å (JCPDS n°. 01-080-2377 standard hematite powder diffraction pattern), as previously predicted by Ayachi et al. [64]. Moreover, the peaks observed at 2θ (°) = 30.07, 35.152, 43.04, 47.12, 53.391, 62.50, 65.71 and 66.50 may be assigned to the crystallographic reflection planes (220), (311), (400), (331), (422), (440), (531) and (442), respectively, corresponding to the crystalline cubic structure of magnetite (Fe_3_O_4_); space group: Fd3m N°: 227 with standard crystallographic parameter a, b and c = 8.3840 Å (JCPDS n°. 01-075-0033 standard magnetite powder diffraction pattern). These results are consistent with those obtained by Noukelag et al. [65], who synthesized hematite nanoparticles using Rosmarinus leaf extract, and Venkateswarlu et al. [66], who synthesized Fe_3_O_4_-NPs from Syzygium cumini seed extract. Other diffraction peaks of lower intensity were observed at 2θ (°) = 28.49 and 41.56, which may be attributed to crystallographic reflection planes (008) and (324), respectively, which correspond to the crystalline tetragonal structure of deviated magnetite (Fe_3-δ_O_4_); space group: P41212 N°: 92 with standard crystallographic parameter a = b = 8.35 Å and c = 25.04 Å (JCPDS n°. 01-080-2186 standard iron oxide powder diffraction pattern) [67]. The CS Fe_x_O_y_-NPs presented additional peaks at 2θ (°) = 27.43, 31.68, 45.43, 55.52 and 65.39, which may be attributed to crystallographic reflection planes (211), (220), (332), (440) and (622), corresponding to the crystalline cubic structure of iron oxide (Fe_2_O_3_); space group: Ia3 N°: 206 with standard crystallographic parameter a = 9.404 Å (JCPDS n°. 00-039-0238 standard iron oxide powder diffraction pattern). Furthermore, peaks were observed at 2θ (°) = 31.68, 44.16, 56.52 and 66.29, which may be attributed to crystallographic reflection planes (314), (440), (643) and (531), corresponding to the crystalline orthorhombic structure of magnetite (Fe_3_O_4_); space group: Pmc21 N°: 26 with standard crystallographic parameter a = 11.868 Å, b = 11.851 Å and c = 16.752 Å (JCPDS n°. 01-076-0957 standard iron oxide powder diffraction pattern) [68].

The parameters obtained from the XRD profiles are shown in Table 2. The proportions of Fe_2_O_3_ and Fe_3_O_4_ in the NPs were obtained by X-ray powder pattern analysis (PDF4) by means of X’Pert HighScore Plus software [69]. A high proportion of Fe_3_O_4_ in the NPs was found in the presence of polyphenols. This may be related to the increased ability of phenolic compounds to reduce Fe^+3^ to Fe^+2^ (Fe_3_O_4_) according to Equation (6) [70]:3Fe^+3^(aq) + 8 − OH^−^ → Fe_3_O_4_ ↓ + 4H_2_O(6)

However, the proportion of Fe_3_O_4_ decreased when the drying temperature increased. A similar effect was related to the decrease in TPC with increasing temperature, as there were not enough polyphenols to reduce Fe^3+^ to Fe^2+^ [46], which may lead to magnetite oxidation according to Equation (7) [70]:4Fe_3_O_4_ + O_2_ → 6Fe_2_O_3_(7)

However, this effect cannot be explained solely by the reduction in TPC. Thus, the proportion of Fe_3_O_4_ achieved at the highest drying temperature is much lower than that reported for the CS NPs, although the absence of some peaks in the XRD profile of CS Fe_x_O_y_-NPs indicates the deficiency of the reducing agent [4].

In contrast, the results suggest that a higher drying temperature inhibits the reduction of Fe^3+^ to Fe^2+^. With respect to the crystal size, drying temperature seems to proportionally affect the nanoparticle size. Thus, the higher the drying temperature, the larger the nanoparticles obtained. The extracting solvent also affects nanoparticle size, although more gently. These results could confirm the relevance of the role of polyphenols, as increased TPC results in a smaller hydrodynamic system, affording smaller nanoparticles [71,72]. In this way, water gave rise to smaller nanoparticles than ethanol, whereas the largest sizes corresponded to the CS Fe_x_O_y_-NPs.

With respect to crystallinity, the well-defined and sharper peaks in the XRD profiles support the good crystallinity degree obtained for the different GS Fe_x_O_y_-NPs. All the GS samples showed a crystalline degree more significant than that of CS Fe_x_O_y_-NPs, increasing with the drying temperature. The lower crystallinity obtained through chemical synthesis is probably due to the lower availability of stabilizing agents [4]. 

#### 3.2.2. Fourier Transform Infrared Spectroscopy (FTIR)

The FTIR profile of the different GS and CS Fe_x_O_y_-NPs were compared, as shown in Figure 4. In addition, they were related to those obtained for raw powder (Appendix A). Three distinct zones could be distinguished: the bonds of Fe-O in iron oxide (800–400 cm^−1^), the oleate’s COO^-^ group (1800–900 cm^−1^) and the chains of the alkyl surface (3000–2800 cm^−1^) [56].

The first observation was focused on the range of 800–400 cm^−1^, where different Fe-O bands can be identified as both maghemite/hematite g-Fe_2_O_3_/α-Fe_2_O_3_ and magnetite Fe_3_O_4_. A powder consisting solely of magnetite has a spectrum with a single band located at 590–580 cm^−1^, unlike maghemite, which has several bands very close to each other in the range of 800–400 cm^−1^, the resolution of which depends on the structural order (previous study) [56].

On the other hand, the surface oxidation of magnetite (Fe_3_O_4_) is represented at 575 cm^−1^, with a maximum peak, followed by a shoulder around 700 cm^−1^. The maghemite spectrum is more complex, with six bands in the range of 800–500 cm^−1^ (the most intense band at 630 cm^−1^) [8,50]. Casillas et al. [73] reported that the magnetite spectra show a characteristic band at approximately 590 and 450 cm^−1^ due to the Fe-O bond in the tetrahedral and octahedral positions, mentioning that the band at 600 cm^−1^ broadens when the particle size decreases. The peaks observed between 630 and 580 cm^−1^ could be assigned to the stretching vibration of the Fe-O bond in the crystalline lattice of magnetite (Fe_3_O_4_). Furthermore, although the literature shows no bonds above 600 cm^−1^ [74], new absorption bands with noticeable intensity located around 1627 and 1390 cm^−1^, in addition to other peaks at 1285 and 1085 cm^−1^ with lower intensity, appear in the Fe_3_O_4_ FTIR spectra [75]. In addition, the stretching vibration of Fe-O of the tetrahedral iron atom in Fe_3_O_4_ appears around 620 cm^−1^ [76]. However, the presence of peculiar bonds in the range of 740–620 cm^−1^ could be attributed to the maghemite phase (γ-Fe_2_O_3_) [74].

The strong characteristic bands at 465 and 559 cm^−1^ suggest a hematite phase, which could correspond to the stretching vibration of Fe-O. The appearance of a signal at 1136 cm^−1^ indicates the presence of the α-Fe_2_O_3_ phase attributed to crystalline Fe-O vibration [65,77]. Absorption peaks were observed at 541 and 566 cm^−1^ for hematite (99.99% and 96% purity, respectively, compared with 100% hematite), which presented an absorption band at 540 cm^−1^, making it possible to confirm and quantify the oxide purity [78].

The weak bands observed around 687 cm^−1^ could indicate magnetite oxidation to maghemite during synthesis [76]. The shoulder band at 635 cm^−1^ is attributed to maghemite (γ-Fe_2_O_3_), and shift from around 600 cm^−1^ to 560 and 475 cm^−1^ indicates the formation of hematite (α-Fe_2_O_3_) [79]. According to this analysis, magnetite was converted to γ-Fe_2_O_3_ and α-Fe_2_O_3_ [79]. These bands were more pronounced in the samples dried at high temperatures. Furthermore, the bands observed around 586 cm^−1^ may be attributed to the reduction of α-Fe_2_O_3_ to Fe_3_O_4_, as they are more pronounced in the samples dried at low temperatures [80]. In this way, the polyphenol compounds found in the FTIR profile of the extracts decreased in the nanoparticle profiles, appearing in new peaks corresponding to Fe_x_O_y_ interactions. The reduction of Fe^3+^ (Fe_2_O_3_) to Fe^2+^ (Fe_3_O_4_) occurs mostly as a result of phenolic compounds found in extracts [46].

Regarding the raw materials, *Phoenix dactylifera* L. can be determined thanks to the vibrational stretching of OH, which is present in the polyphenol groups [81]. These bonds are observed with a band at 3416 cm^−1^, which is also seen in some NPs. Two sharp peaks appeared between 2913 and 2839 cm^−1^, which could be related to the extension mode of hydrocarbon. In addition, other bands can be associated with different bonds. The band at 1630 cm^1^ could be related to the aromatic ring deformation or C=C stretching vibration of alkane groups. The band found around 1735 cm^−1^ could be assigned to the C=O bonds of aldehydes, ketones and ester. The bands around 1380 are associated with ester groups [46], [82]. The C-O asymmetric stretching vibration that is typical in polyphenol compounds is observed between 1200 and 1247 cm^−1^ [46]. The C-O-C stretching vibration of phenolic compounds is also seen between 1039 and 1070 cm^−1^ [46]. Finally, the bonds between 1105 and 1160 are assigned to C–O–H in phenolic compounds [46]. Comparing the spectra of polyphenol extracts and NPs, a division of the 1643 cm^−1^ band into three different peaks (1653, 1633 and 1623 cm^−1^) is observed. This behavior is caused by the reduction of FeCl_3·_·6H_2_O with the oxygen atoms of phenolic groups (–OH).

#### 3.2.3. Scanning Electron Microscopy (SEM)

The SEM micrographs of both GS and CS Fe_x_O_y_-NPs are shown in Figure 5. The GS Fe_x_O_y_-NPs exhibited different clustered nanostructures with spherical morphology and face-centered rhombohedral, cubic and hexagonal structures, with a more homogenous size distribution with slight agglomeration due to the interactions among nanoparticles [46]. Visual analysis suggests that the GS-NPs prepared at lower drying temperatures tend to form slight agglomerates or aggregates, either as a result of phenolic compounds reacting with the surfaces of Fe_x_O_y_-NPs or due to the biological compounds present on the particles’ surfaces. The occurrence of H bonding in bioactive molecules could lead to the aggregation of nanoparticles [54,83,84]. The slight-to-null agglomerations in green synthesis of NPs could be explained by the aging effect under reflux conditions and phytochemicals present in Phoenix extracts ruling as stabilizing agents [4]. Moreover, the CS Fe_x_O_y_-NPs had a greater tendency to be more agglomerated, which is probably due to the unavailability of stabilizing agents [4].

Figure 6 shows the nanoparticle size distribution, with an average diameter of 35 ± 1 to 44 ± 1 nm for the GS Fe_x_O_y_-NPs and 59 ± 20 nm for the CS Fe_x_O_y_-NPs. These results are in substantial agreement with the XRD results, leading to similar conclusions.

#### 3.2.4. Antioxidant Activity

Table 2 summarizes the DPPH *IC_50_* values for the different GS Fe_x_O_y_-NPs and GS Fe_x_O_y_-NPs. Unlike in the case of gallic acid, a lower *IC_50_* value is associated higher antioxidant activity. These values could illustrate the effect of primary factors on certain substances of extracts, such as the polyphenol content, the synthesis of their nanoparticles and, finally, their antioxidant activity.

The obtained results indicate that the highest activity was attributed to the nanoparticles synthesized from the extracts prepared from the leaves dried at lower temperatures. In this way, GS Fe_x_O_y_-NPs-T_25 °C (H2O Ext or Ethanol Ext.)_ produced higher scavenging activity than the NPs synthesized from leaves dried at higher temperatures, CS NPs-Fe_x_O_y_ and the extracts themselves, including the standard gallic acid (*IC_50_* = 0.41 ± 0.20 mg/mL). This could be explained by the simultaneous activity of polyphenols remaining as antioxidant agents and GS NPs-Fe_x_O_y_ as a catalyst [85]. In addition, it was found that aqueous extracts had higher polyphenol content than ethanolic extracts. This phenomenon is likely due to the presence of phytochemicals and iron ions, which may act as antioxidants by transferring single electrons and hydrogen atoms [86] or by releasing oxygen atoms [4]. Other studies have reported the adsorption of bioactive compounds of extracts on spherical nanoparticles [87]. Similar results were reported by Zdenka et al. [88], who investigated the antioxidant activity of green and chemical Ag-NPs.

Nevertheless, the GS Fe_x_O_y_-NPs-T _50 °C H2O Ext_ exhibited lower antioxidant activity compared with NPs obtained with ethanol (Fe_x_O_y_-NPs-T _50 °C Ethanol Ext_), the raw extract and the standard extract (gallic acid) [89], which can be attributed to Fe_3_O_4_-NPs % present in the sample. Patra et al. reported that this may be due to the stereoselectivity of the bioactive complex present on the Fe_3_O_4_-NPs surface when acting as an anti-free radical agent [90]. In addition, the degradation of polyphenolic compounds in hot water upon prolonged exposure to high temperatures may also affect antioxidant activity, resulting in a decrease in radical scavenging activity [41].

Table 2 shows the relative *IC_50 NPs_/IC_50 Ext_*. Values lower than 1 express higher scavenging activity in Fe_x_O_y_-NPs than the extracts, and values of more than 1 express the opposite. In this case, all extracts presented higher scavenging activity than NPs. These results make sense, considering that part of the antioxidant activity of the extract is lost to reduce FeCl_3_·6H_2_O and synthesize the NPs. Furthermore, from a nanoscale point of view, it was observed that when the iron-reducing capacity increases, the hydrodynamic sizes of the NPs-Fe_x_O_y_ tend to decrease [46].

Table 2 shows the TAC values of GS Fe_x_O_y_-NPs and CS Fe_x_O_y_-NPs. The highest total antioxidant activity values were produced for the green nanoparticles synthesized from the ethanolic extract, especially at lower drying temperatures. However, as shown in Table 1, the highest TAC value was achieved by T_25 °C H2O Ext._, with 2.04 mg GAE/mg extract. This means that although the NPs synthesized from ethanolic extracts have more total antioxidant groups, they do not exhibit a good anti-radical function (higher DPPH *IC_50_* values).

## 4. Conclusions

The synthesis of iron oxide nanoparticles using *Phoenix dactylifera* L. has been shown to be an alternative and feasible method that is environmentally safer than conventional chemical methods, in addition to being harmless to human health.

Nevertheless, the phenolic compounds present in *Phoenix dactylifera* L. could be affected by various factors, especially the drying temperature of the plant and the extraction solvent used, producing differences in the final nanoparticles obtained. Therefore, increasing the drying temperature could affect the phenolic compound efficiency (reducing capacity) and, subsequently, the nanoparticle size, crystallinity and the type of oxide obtained. The green nanoparticles obtained at lower drying temperatures exhibited the best physicochemical properties and antioxidant activity throughout the process.

Finally, nanoparticle size and morphology were decisive in their unique properties, as they can affect antioxidant activity.

## Figures and Tables

**Figure 1 nanomaterials-12-02449-f001:**
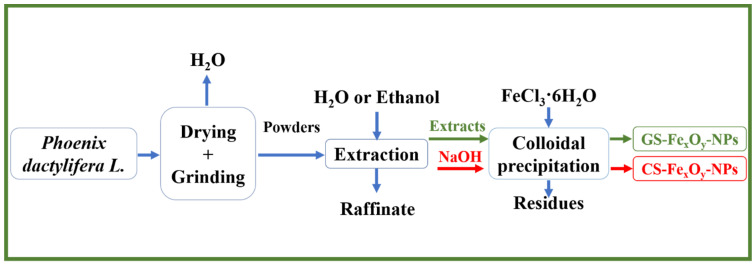
Diagram of the synthesis of GS Fe_x_O_y_-NPs (green) and CS Fe_x_O_y_-NPs (red).

**Figure 2 nanomaterials-12-02449-f002:**
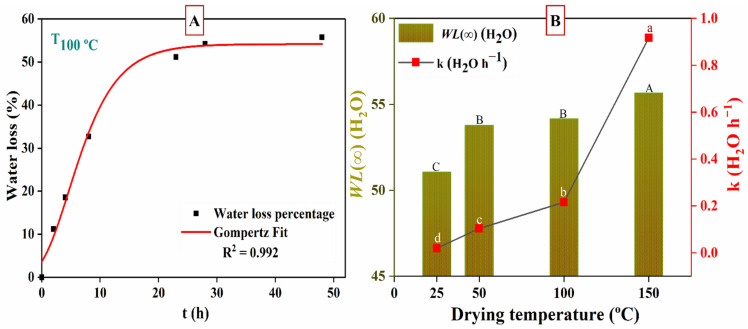
(**A**) Water loss as a function of drying time (t) at T_100 °C_. (**B**) Maximum water loss percentage, WL(∞), and kinetic parameter (k) as a function of drying temperature.

**Figure 3 nanomaterials-12-02449-f003:**
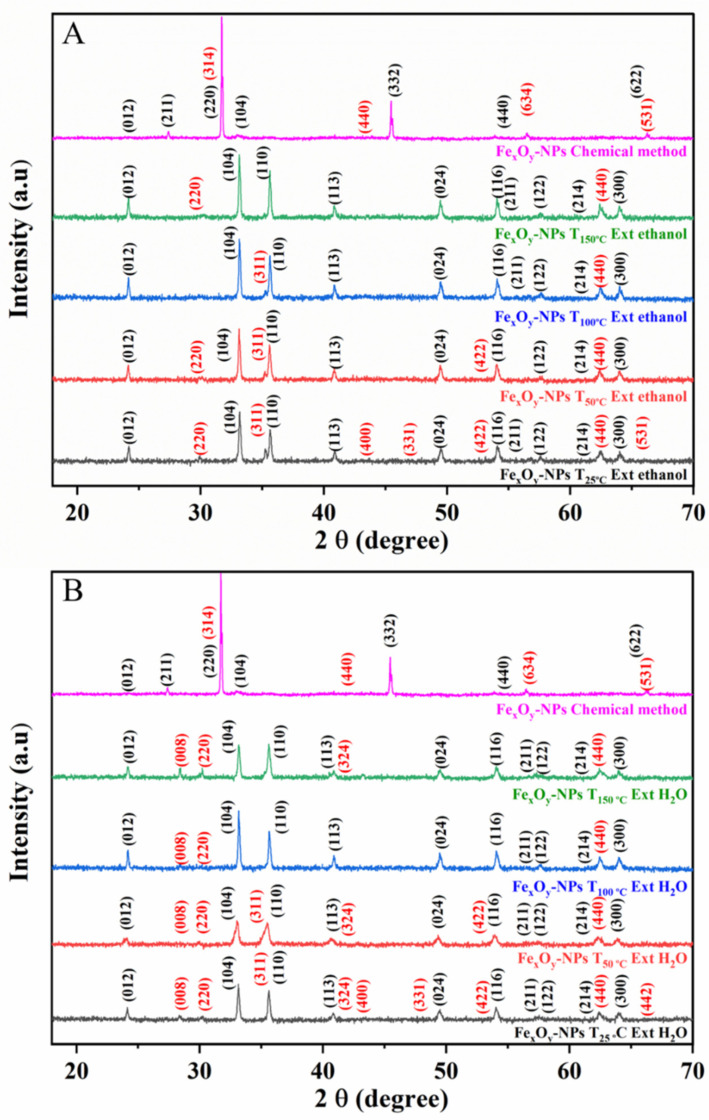
XRD spectra of the GS Fe_x_O_y_-NPs (JCPDS standard) using ethanolic (**A**) and aqueous (**B**) extracts obtained at different drying temperatures. The profile CS Fe_x_O_y_-NPs is included for comparison.

**Figure 4 nanomaterials-12-02449-f004:**
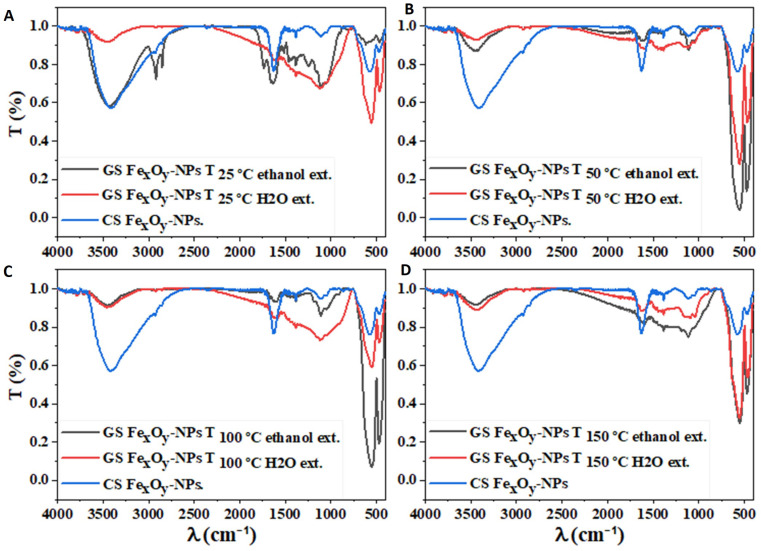
FTIR spectra of the GS Fe_x_O_y_-NPs using ethanolic or aqueous extracts obtained at different drying temperatures: 25 °C (**A**), 50 °C (**B**), 100 °C (**C**) and 150 °C (**D**). The profile of CS Fe_x_O_y_-NPs is included for comparison.

**Figure 5 nanomaterials-12-02449-f005:**
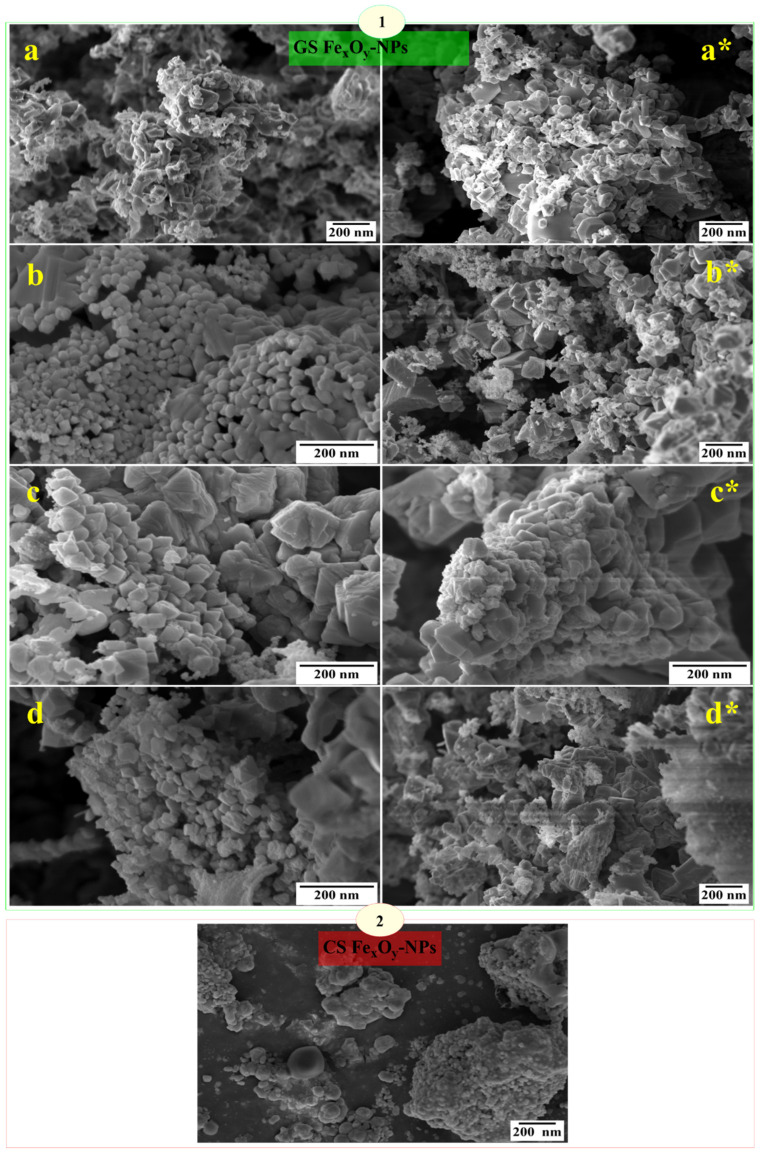
Scanning electron microscopy (SEM) images of (1) GS Fe_x_O_y_-NPs using ethanolic extracts: (**a**) T_25 °C_, (**b**) T _50 °C_, (**c**) T _100 °C_ and (**d**) T _150 °C_ and aqueous extracts: (**a***) T_25 °C_, (**b***) T _50 °C_, (**c***) T _100 °C_ and (**d***) T _150 °C_ and (2) CS Fe_x_O_y_-NPs.

**Figure 6 nanomaterials-12-02449-f006:**
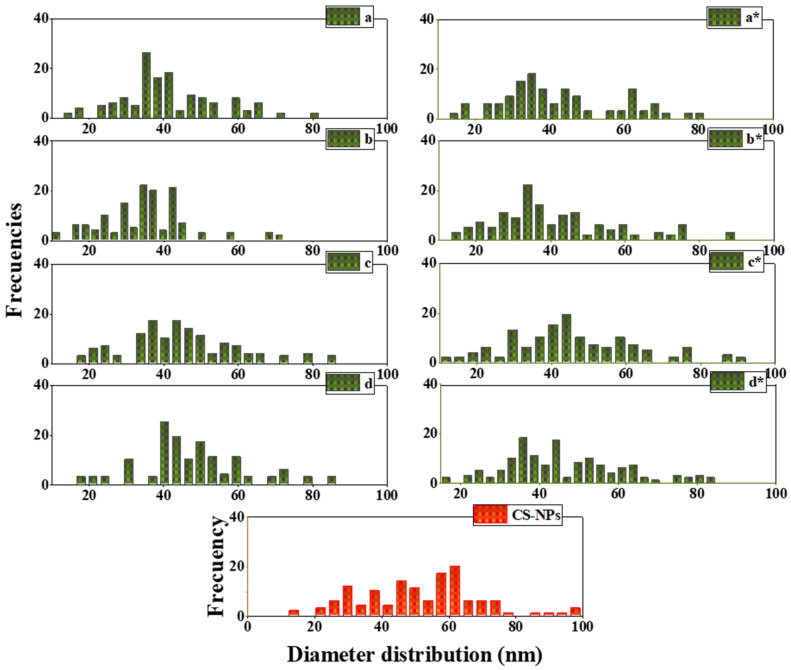
Particle size distribution of GS Fe_x_O_y_-NPs using ethanolic extracts: (**a**) T_25 °C_, (**b**) T _50 °C_, (**c**) T _100 °C_ and (**d**) T _150 °C_ and aqueous extracts: (**a***) T_25 °C_, (**b***) T _50 °C_, (**c***) T _100 °C_ and (**d***) T _150 °C_. The profile of CS Fe_x_O_y_-NPs is included for comparison.

**Table 1 nanomaterials-12-02449-t001:** Results of the average particle size of obtained powders, the extraction yield, total polyphenol content (TPC) of the extracts, IC50 DPPH free radical and total antioxidant activity (TAC) as a function of drying temperatures (25, 50, 100 and 150 °C) and solvents (ethanol and H_2_O). Different superscript letters (a–d) within a column indicate significant differences among mean observations (*p* < 0.05).

Sample	Size (µm)	Extraction Yield (%)	TPC (mg GAE/g Extract)	TAC (mg GAE/mg Extract)
T_25_ _°C_ _Etanol Ext._	90.56 ± 2.35 ^a^	53.77 ± 0.77 ^c^	38.64 ± 1.72 ^a,b^	0.92 ± 0.45 ^a,b^
T_25_ _°C_ _H2O Ext._		51.35 ± 0.57 ^d^	42.28 ± 0.79 ^a^	2.04 ± 0.48 ^a^
T_50_ _°C_ _Etanol Ext._	47.75 ± 1.48 ^b^	56.13 ± 0.29 ^b^	35.13 ± 3.17 ^b,c^	0.87 ± 0.21 ^a,b^
T_50_ _°C_ _H2O Ext._		54.45 ± 0.77 ^b,c^	38.47 ± 2.01 ^a,b^	1.95 ± 0.75 ^a,b^
T_100_ _°C_ _Etanol Ext._	44.85 ± 1.25 ^b^	58.53 ± 0.99 ^a^	33.23 ± 2.72 ^b,c,d^	0.77 ± 0.52 ^a,b^
T_100_ _°C_ _H2O Ext._		55.15 ± 0.72 ^b,c^	37.55 ± 2.04 ^a,b^	1.70 ± 0.28 ^a,b^
T_150_ _°C_ _Etanol Ext._	36.98 ± 1.07 ^a^	59.63 ± 0.65 ^a^	29.02 ± 1.50 ^d^	0.66 ± 0.30 ^b^
T_150_ _°C_ _H2O Ext._		59.57 ± 0.77 ^a^	29.59 ± 0.85 ^d,c^	1.49 ± 0.63 ^a,b^

**Table 2 nanomaterials-12-02449-t002:** Results obtained for different parameters of GS Fe_x_O_y_-NPs as a function of drying temperature (25, 50, 100 and 150 °C) and solvent (ethanol and H_2_O) and CS Fe_x_O_y_-NPs: composition (proportion of Fe_2_O_3_ and Fe_3_O_4_), crystallinity percentage, average diameter in nm (from DRX measurements and SEM images) and antioxidant activity (*IC_50_* DPPH free radical in mg/mL, the ratio between *IC_50_* of NPs and extracts, *IC_50 NPs_/IC_50 Ext_* and total antioxidant activity (TAC) in mg GAE/mg). Different superscript letters (a–d) within a column indicate significant differences among mean observations (*p* < 0.05).

Sample		Fe_2_O_3_ (%)	Fe_3_O_4_ (%)	Crystallinity (%)	D _DRX_ (nm)	D _SEM_ (nm	*IC_50_ *(mg/mL)	*IC_50 NPs_/IC_50 Ext_*	TAC (mg GAE/mg)
GS Fe_x_O_y_-NPs	T _25 °C_ _Ethanol Ext._	45.8	54.2	68.7 ^g^	36 ± 10	37 ± 1 ^d^	0.29 ± 0.11 ^b^	0.55	24.76 ± 1.48 ^a^
	T _25 °C_ _H2O Ext__._	46.3	53.7	66.5 ^h^	36 ± 12	36 ± 1 ^d^	0.25 ± 0.09 ^b^	0.54	18.34 ± 1.09 ^a,b,c^
	T _50 °C Ethanol Ext__._	79.8	20.2	71.3 ^e^	38 ± 10	38 ± 1 ^c,d^	0.64 ± 0.31 ^a,b^	0.96	21.54 ± 1.70 ^a,b^
	T _50 °C_ _H2O Ext._	75.1	24.9	71.0 ^f^	38 ± 7	37 ± 1 ^d^	1.39 ± 0.67 ^a,b^	2.14	15.82 ± 0.70 ^b,c^
	T _100 °C_ _Ethanol Ext__._	87.6	12.4	76.0 ^c^	42 ± 16	41 ± 2 ^b,c^	1.14 ± 0.90 ^a,b^	2.58	21.41 ± 0.29 ^a,b^
	T _100 °C H2O Ext._	88.9	11.1	73.8 ^d^	42 ± 16	42 ± 1 ^b,c^	0.68 ± 0.41 ^a,b^	1.66	12.43 ± 2.20 ^c,d^
	T _150 °C_ _Ethanol Ext__._	90.2	9.8	85.9 ^b^	44 ± 16	45 ± 1 ^b^	2.12 ± 1.64 ^a^	4.07	15.31 ± 0.78 ^b,c^
	T _150 °C_ _H2O Ext._	92.8	7.2	86.6 ^a^	45 ± 16	43 ± 1 ^b^	0.81 ± 0.10 ^a,b^	0.92	12.51 ± 3.46 ^c,d^
CS Fe_x_O_y_-NPs		68.9	31.31	46.7 ^i^	59 ± 24	59 ± 20 ^a^	1.24 ± 0.32 ^a,b^	-	8.56 ± 0.84 ^d^

## Data Availability

The data presented in this study are available on request from the corresponding author.

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
