# Peer review of "Green Synthesis of FexOy Nanoparticles with Potential Antioxidant Properties"

_nanomaterials, 2022, doi:10.3390/nano12142449_

Round 1

Reviewer 1 Report

The paper is of high methodological importance, and object of study is of high actuality.

However, corrections are necessary:

1) The Article uses a lot of abbreviations, it misleads readers. So I advise to include a list of abbreviation at the beginning or end of the text.  

2) Language correction by native speaker is necessary, with special attention to Abstract and Introduction. Some misleads and mistakes are presented below:

line 18: , using iron chloride was used as precursor.

line 36: Therefore, researchers have been recently investigating the use of environmentally

88-89: The comparison was based on their physicochemical characteristics and their antioxidant activity (TAC and DPPH).

 3) Two identical titles of subsection: 3.1.1.  and  3.1.2. (lines 211 and 232):  Effect of drying temperature on moisture content in Phoenix dactylifera L.

4) Please present descriptions that are more exact:

4.1).  in conclusion p. 479: The green nanoparticles obtained from lower drying temperatures were the best in the whole process. (“the best” – for what conditions?)

4.2) p. 482: their antioxidant and biological activity (“antioxidant” activity involves biological aspect).

Author Response

Dear reviewer,

  Thank you for your comments. Attached you can find the response to your comments. 

Kind regards,

Alberto Romero

Reviewer 2 Report

(1) English in the paper should be carefully revised.

(2) Pay attention to the subscript and abbreviation.

(3) The novelty of paper should be further confirmed, especially for the keyword “Green”, and some references in the related fields are suggested to add. For example, Journal of Advanced Ceramics, 2021, 10 (1): 62-77; Journal of the European Ceramic Society 42 (2022) 4759-4769.

(4) The title is suggested to revise, especially for the “Physicochemical and  functional properties”. 

Author Response

(The authors gave the same response as above.)

Reviewer 3 Report

This manuscript reported a green synthesis of Iron oxide nanoparticles (FexOy-NPs) with Phoenix dactylifera L. extracts. The authors systematically investigated the synthesis condition and compared the physicochemical characteristics and antioxidant activity of green FexOy-NPs and chemical FexOy-NPs. However, plants are commonly used for FexOy-NPs synthesis (Langmuir 2014, 30, 5982–5988; Journal of Environmental Health Science & Engineering, 2015, 13, 1-7; Journal of Magnetism & Magnetic Materials, 2010, 322, 2938-2943.). And many details should be chiefly explained and revised. Therefore, we give you major revision before publication. Comments are listed below.

Other comments: 

1. The manuscript requires revision with respect to the language used. The author should ask a native English speaker or equivalent to assist with correcting the spelling, grammar, word use, and punctuation throughout the manuscript.

2. There are already some reports on the green synthesis of FexOy. The authors need to think about the novelties of this work and re-organized the structure of the manuscript to highlight the novelty.

3. The font size of Figure 2a should be in accordance with that of Figure 2b.

4. The ticks need to be added in figure 4 to mark the temperature label and λ label. 

5. There are some mistakes in References. In the reference [68], specific page numbers should be given. In the reference [61], the year is not in the correct format. In the reference [14], [33], [38] and [58], the DOI should be given. The authors should double-check the manuscript.

6.Some relevant works on synthesis of nanoparticles needs to be cited (Chinese Journal of Chemistry, 2021, 39, 1009; Journal of Rare Earths, 2022, doi.org/10.1016/j.jre.2022.05.016; Nanoscale Advances, 2020; 2: 1380.)

Author Response

(The authors gave the same response as above.)

Round 2

Reviewer 3 Report

The authors have addressed most questions I have raised. In order to clarify the novelty of their work, I suggest the authors to further strengthen the difference of their work compared with previously reported works. In addition, the mechanism underlying the green synthesis process is also needed. There are some grammar and spelling mistakes in the manuscript. The authors should read the whole manuscript carefully and revised them.

Author Response

Once again, the authors would like to extend their gratitude to the reviewer for his/her helpful comments. Accordingly, a novelty part has been included (Lines 84-96), as well as some suggested reaction mechanisms (Lines 320-325). Finally,  orthographic errors in the whole manuscript have been fixed and marked.
